# Access to medicines among the Brazilian population based on the 2019 National Health Survey

**Adriana Amorim de Farias Leal**[1]*, **Maria Helena Rodrigues Galvão**[1], **Arthur de Almeida Medeiros**[1,2], **Ângelo Giuseppe Roncalli**[1]

1 Graduate Program in Public Health, Federal University of Rio Grande do Norte, Natal, Rio Grande do Norte, Brazil, 2 Health Integrated Institute, Federal University of Mato Grosso do Sul, Campo Grande, Mato Grosso do Sul, Brazil

* aafl.cg@gmail.com

## Abstract

### Introduction

Access to medicines is a challenge, especially in developing countries, highlighting the need of population-based research to evaluate access and related factors.

### Objective

This study aimed to assess access to medicines and identify associated factors using data from the 2019 Brazilian National Health Survey (PNS).

### Methods

This population-based cross-sectional study used data from the 2019 PNS and considered access to prescription medicines as the primary outcome. The sample included 24,753 individuals aged 15 years or older who looked for medical care in the last 15 days and received a medicine prescription. Andersen's behavioral model was used to select independent variables. After descriptive analysis, a multinomial logistic regression multilevel analysis was performed using the independent variables with a significance level lower than 0.20 in the bivariate analysis.

### Results

The lowest chances of getting access to medicines were observed in individuals aged between 40 and 59 years, women, with complete middle and high school, with lower-income families, who attended public services, with worse self-assessed health, and those who looked for health care for disease prevention and health promotion.

### Conclusions

Access to medicines among the Brazilian population is associated with social, economic, and health perception factors. Our findings may update and guide the development of public

**Data Availability Statement:** All data files are held in a public repository and are available from the Brazilian Institute of Geography and Statistics (microdata spreadsheets from the 2019 National

Health Survey) database (access from URL: https://www.ibge.gov.br/estatisticas/sociais/saude/9160-pesquisa-nacional-de-saude.html?=&t=resultados).

**Funding:** The authors received no specific funding for this work.

**Competing interests:** The authors have declared that no competing interests exist.

policies on medication and pharmaceutical care, facilitating medication purchases by the care user and promoting health equity.

## Introduction

A quarter of worldwide healthcare cost is on medicines. Still, in many countries, the main source of medicines affordance comes from individuals and families, impairing equality and efficiency [1]. In Brazil, the health care model, including pharmaceutical care and access to medication, is made up of three systems: the universal public system, Unified Health System (SUS), with completely free access; the supplementary health system, which are private plans and insurance; and the "out of pocket" system, in which the service provider is paid directly [2].

Medicines are the main health expense among Brazilian families, possibly leading to a financial burden, especially for families with less purchasing power [3]. Studies indicate that the private costs of medicines are higher in countries without a public health system and that do not follow universal health systems, such as the United States of America, contrary to most European countries [2]. Therefore, access to medication depends on the organization of a health system. Recent studies [4–7] raise the issue of access to medicines in Brazil and other countries, mostly because medicine availability in developing countries is still low.

In 2002, the World Health Organization defined essential medicines as the most effective medications for the healthcare needs of a population. The concept (effective, safe, affordable, and efficient medicines to treat with quality) must be internalized in pharmaceutical care in its integrality [8]. Guaranteeing access to medicines is needed to ensure integrality. In Brazil, integrality is secured by the National Medicines Policy [9], which aims to access medicines, quality control, and rational use. Access to medicines for Farias and collaborators [10] comprises the availability of essential medicines when and where the user requires them, quality control, and enough information for proper medication.

Published in the same year of the National Medicines Policy, the National Research of Household Sample included access to medication as a supplement to access, use, and consumption of health services [11]. Since 2013, the Brazilian National Health Survey (PNS) has conducted the most robust Brazilian population-based research on health issues, including data on access to medicines from public or private systems within the item "access and use of health services" [12].

Therefore, the present study aimed to assess access to medicines and identify individual associated factors using data from the 2019 PNS.

## Materials and methods

### Study design

This population-based cross-sectional study analyzed data from the PNS, a national survey conducted by the Ministry of Health and the Brazilian Institute of Geography and Statistics. The PNS provides a representative database of the country and private households. We used data referring to the second and updated edition of the 2019 PNS, which used a master sample for household sample surveys for greater coverage.

### Theoretical model of the study and sampling

The study characterization was based on item J (entitled "use of health services") of the 2019 PNS. Access to medicines was the dependent variable, while associated factors were set as

independent variables. We adopted individual associated factors (predisposing, enabling, and needed) according to the model suggested by Andersen Behavior Model [13].

Data from the 2019 PNS were collected between August 2019 and March 2020 [12]. It used three stages of cluster sampling: selection of census tracts from primary sampling units, selection of households in each Emergency Care Unit, and selection of residents aged $\geq$ 15 years from each household. From 108,457 selected households, 100,541 were occupied. The database consisted of responses from 279,382 individuals, referring to 94,114 household interviews.

One resident (aged $\geq$ 18 years) of each household selected during the second sampling stage responded to the second part of the survey with questions about all household residents.

We included responses of individuals who sought medical care in the last 15 days but only got it in the previous attempt (question J021 - "The last time you sought health care for this reason, in the last two weeks, was ___ seen?") (40,357 individuals), and answered "yes" regarding having received a medicine prescription (question J29a - "Was any medicine prescribed in this consultation for ___?") (24,753 responders).

## Characterization of variables

**Dependent variable.** The primary outcome was access to prescription medicines. Only affirmative answers to the question "Were you able to obtain the prescription medicines?" were considered.

**Independent variables.** We adopted Andersen Behavior Model [13] of access to health services to select independent variables, considering the individual factors (predisposing, enabling, and needed) associated with access to medicines.

Age (0–14; 15–19; 20–39; 40–59; $\geq$ 60 years), sex (men or women), and level of education (no schooling, incomplete or complete middle school, complete high school, and higher education) were considered individual predisposing factors.

Enabling factors included family income, calculated using the purchasing power parities index [14]. This index compares cost of living and income among different countries, measured in national currency by US dollar (up to $124, from $125 to $248, or $249 or more). Enabling factors also included the nature of care provided in the last 15 days (public or private) and whether SUS provided it or the individual or the healthcare insurance paid for it.

Needed factors included general health status (very good or good; regular; bad or very bad), the perceived need or reason to seek health care (disease diagnosis and treatment, disease prevention and health promotion, consultations and procedures, and external causes and rehabilitation), and the assessed need or type of health care provided (disease prevention and health promotion, consultations and procedures, medium or high complexity service, and others).

## Ethics statement

This study used secondary data (PNS), which were public and available on the Brazilian Institute of Geography and Statistics website (www.ibge.gov.br). The 2019 PNS met all requirements for research involving human beings and was approved by the national research ethics committee (protocol n˚ 3.529.376).

## Statistics

Due to the complex sample design, the descriptive analysis required a population expansion considering the sample weight and the cluster effect. We set the estimated prevalence at 95% confidence interval (95% CI) for all variables. Then, we conducted a bivariate analysis crossing dependent and independent variables.

The degrees of association between access to medicines and independent variables were assessed using multinomial logistic regression. This analysis model was applied only to the variables with a significance level lower than 0.20 in the bivariate analysis. Estimates were interpreted by odds ratio (OR) with 95% CI. The analysis of the fit of the final model was performed using the likelihood ratio test and Pseudo $R^2$. Data were analyzed using SPSS software (version 20.0; IBM Corp., NY, USA).

## Results

According to the 2019 PNS, 108,457 households were visited with a response rate of 93.6% (n = 293,725). Regarding the demand for health care in the last 15 days, only 18.6% of the interviewed individuals sought medical care, of which 86.1% received it. The highest proportion of medical appointments was in SUS (57.2%). Two in every three appointments had medicine prescriptions (64.8%).

Regarding individual predisposing factors, for those with access to the prescribed medication, 86.4% were between 0 and 14 years, 86.4% were men, and 89.3% had more education. In enabling factors, 90.3% of individuals with access to prescription medicines had an income higher than U$ 249.00, and 90.8% received private health care in the last 15 days. In contrast, 90.2% of those with no access to prescription medicines received care via SUS. For needed factors, 88.5% of those with access to prescription medicines rated their general health status as very good or good (Table 1).

Table 2 presents the raw analysis of access to prescription medicines among the Brazilian population. For predisposing factors, adults between 40 and 59 years had a higher probability of having partial or no access to prescription medicines than older individuals. Women also had a higher probability of partial, total, or no access than men. Regarding the level of education, individuals without schooling were more likely to have partial access to prescription medicines. In contrast, individuals with complete middle school were more likely to have no access to prescription medicines than those with higher education. For enabling factors, the probability of partial or no access to prescription medicines was greater in lower-income families. Also, a higher chance of partial or no access to prescription medicines occurred in the public system (SUS) than in the private system. Regarding needed factors, the probability of partial or no access to prescription medicines in the last medical appointment increased with worse health self-assessment, following last medical appointment.

Table 3 presents the adjusted analysis of access to prescription medicines in the Brazilian population using multinomial logistic regression, considering the category access to all prescribed drugs of the dependent variable as the analysis reference. For predisposing factors, adults between 40 and 59 years had a 30% higher probability of having no or partial access to prescription medicines than older individuals ($\geq$ 60). Women were 12% and 18% more likely to have partial and no access to prescription medicines, respectively, than men.

Individuals with complete middle and high school had 26% and 24%, respectively, greater chances of no access to prescription medicines than those with higher education. Regarding enabling factors, we found a high probability of partial or no access to prescription medicines for individuals in lower-income families. Individuals with a family income per capita of up to U$ 249.00 had a 67% probability of partial access and a 56% probability of no access to prescription medicines.

Moreover, individuals who had appointments in the public health system were 2.6-fold more likely to have partial access to prescription medicines and had a 36% higher chance of having no access to prescription medicines than those attending the private system.

**Table 1. Frequency of access to prescription medicines according to the 2019 PNS.**

| | Access to prescription medicines | | | | | |
| --- | --- | --- | --- | --- | --- | --- |
| | Yes, all | | Yes, some | | No, none | |
| | *n* | %<br>(95% CI) | *n* | %<br>(95% CI) | *n* | %<br>(95% CI) |
| *Predisposing factors* | | | | | | |
| **Age (years)** | | | | | | |
| 0–14 | 3,286,302 | 86.4<br>(84.3; 88.3) | 269,740 | 7.1<br>(5.5; 9.0) | 246,396 | 6.5<br>(5.2; 8.0) |
| 15–19 | 844,668 | 83.9<br>(80.3; 86.9) | 68,639 | 6.8<br>(5.0; 9.3) | 93,787 | 9.3<br>(6.9; 12.4) |
| 20–39 | 3,953,771 | 85.3<br>(83.6; 86.8) | 319,887 | 6.9<br>(5.8; 8.2) | 363,446 | 7.8<br>(6.7; 9.1) |
| 40–59 | 5,111,233 | 83.4<br>(82.0; 84.7) | 538,017 | 8.8<br>(7.8; 9.8) | 477,137 | 7.8<br>(6.9; 8.8) |
| $\geq$ 60 | 4,198,525 | 86.1<br>(84.5; 87.5) | 386,864 | 7.9<br>(6.7; 9.3) | 293,764 | 6.0<br>(5.2; 7.0) |
| **Sex** | | | | | | |
| Women | 10,555,018 | 84.2<br>(83.1; 85.3) | 1,029,108 | 8.2<br>(7.4; 9.1) | 948,533 | 7.6<br>(6.9; 8.3) |
| Men | 6,839,481 | 86.4<br>(85.0; 87.5) | 554,038 | 7.0<br>(6.0; 8.1) | 525,996 | 6.6<br>(5.8; 7.6) |
| **Level of education** | | | | | | |
| No schooling | 1,543,956 | 80.6<br>(77.3; 85.3) | 227,496 | 11.9<br>(9.1; 15.3) | 144,758 | 7.6<br>(6.0; 9.5) |
| Incomplete middle school | 5,563,985 | 84.3<br>(82.9; 85.6) | 582,562 | 8.8<br>(7.8; 10.0) | 455,406 | 6.9<br>(6.1; 7.8) |
| Complete middle school | 2,083,888 | 82.4<br>(80.0; 84.6) | 218,877 | 8.7<br>(7.2; 10.4) | 225,394 | 8.9<br>(7.3; 10.8) |
| Complete high school | 4,252,611 | 85.8<br>(84.3; 87.3) | 327,655 | 6.6<br>(5.6; 7.9) | 373,461 | 7.5<br>(6.6; 8.7) |
| Higher education | 2,368,427 | 89.3<br>(87.1; 91.1) | 128,349 | 4.8<br>(3.7; 6.3) | 155,578 | 5.9<br>(4.5; 7.6) |
| *Enabling factors* | | | | | | |
| **Family income per capita** | | | | | | |
| Up to $252.52 | 4,548,364 | 79.3<br>(77.3; 81.2) | 636,965 | 11.1<br>(9.6; 12.9) | 547,474 | 9.6<br>(8.3; 11.0) |
| $252.53 to $547.57 | 5,846,112 | 83.9<br>(82.4; 85.3) | 616,438 | 8.8<br>(7.8; 10.0) | 503,474 | 7.2<br>(6.4; 8.2) |
| $547.58 or more | 6,993,278 | 90.3<br>(89.0; 91.5) | 329,743 | 4.3<br>(3.5; 5.1) | 422,446 | 5.5<br>(4.6; 6.5) |
| **Nature of care** | | | | | | |
| Private | 6,884,762 | 90.8<br>(89.6; 91.8) | 288,220 | 3.8<br>(3.1; 4.6) | 412,315 | 5.4<br>(4.7; 6.3) |
| Public | 10,481,560 | 81.7<br>(80.5; 82.8) | 1,292,551 | 10.1<br>(9.2; 11.0) | 1,061,290 | 8.3<br>(7.5; 9.1) |
| Do not know / Do not remember | 28,176 | 89.5<br>(75.4; 96.0) | 2,375 | 7.5<br>(2.2; 22.5) | 925 | 2.9<br>(1.0; 8.3) |
| **Health insurance** | | | | | | |

(*Continued*)

**Table 1.** (Continued)

| | Access to prescription medicines | | | | | |
|---|---|---|---|---|---|---|
| | Yes, all | | Yes, some | | No, none | |
| | *n* | % (95% CI) | *n* | % (95% CI) | *n* | % (95% CI) |
| Yes | 5,121,575 | 90.7 (89.2; 92.0) | 221,487 | 3.9 (3.1; 4.9) | 302,699 | 5.4 (4.4; 6.5) |
| No | 12,272,924 | 82.9 (81.8; 83.9) | 1,361,659 | 9.2 (8.4; 10.1) | 1,171,830 | 7.9 (7.2; 8.7) |
| **Paid appointment** | | | | | | |
| Yes | 2,891,838 | 89.3 (87.3; 91.0) | 133,763 | 4.1 (3.1; 5.4) | 213,034 | 6.6 (5.2; 8.3) |
| No | 14,502,661 | 84.3 (83.3; 85.2) | 1,449,383 | 8.4 (7.7; 9.2) | 1,261,495 | 7.3 (6.7; 8.0) |
| **Appointment via SUS** | | | | | | |
| Yes | 10,187,286 | 81.8 (80.6; 82.9) | 1,247,622 | 10.0 (9.1; 11.0) | 1,022,287 | 8.2 (7.4; 9.0) |
| No | 7,151,192 | 90.2 (88.9; 91.3) | 331,855 | 4.2 (3.4; 5.2) | 449,314 | 5.7 (4.9; 6.6) |
| Do not know / Do not remember | 56,020 | 89.5 (77.4; 95.5) | 3,670 | 5.9 (1.9; 16.6) | 2,929 | 4.7 (1.2; 16.3) |
| *Needed factors* | | | | | | |
| **General health status** | | | | | | |
| Very good or good | 9,525,744 | 88.5 (87.4; 89.5) | 538,019 | 5.0 (4.2; 5.9) | 704,268 | 6.5 (5.8; 7.4) |
| Regular | 5,879,950 | 83.1 (81.7; 84.4) | 679,452 | 9.6 (8.5; 10.8) | 516,535 | 7.3 (6.5; 8.2) |
| Bad or very bad | 1,989,505 | 76.3 (73.2; 79.1) | 365,675 | 14.0 (11.9; 16.4) | 253,726 | 9.7 (7.6; 12.4) |
| **Reason to seek health care (perceived need)** | | | | | | |
| Disease diagnosis and treatment | 12,112,024 | 85.1 (84.1; 86.1) | 1,146,858 | 8.1 (7.3; 8.9) | 971,229 | 6.8 (6.2; 7.6) |
| Disease prevention and health promotion | 2,110,170 | 85.4 (83.2; 87.4) | 142,755 | 5.8 (4.7; 7.1) | 217,098 | 8.8 (7.2; 10.7) |
| Consultations and procedures | 1,668,093 | 86.5 (84.2; 88.6) | 133,455 | 6.9 (5.3; 9.0) | 125,929 | 6.5 (5.3; 8.0) |
| External causes and rehabilitation | 1,504,211 | 82.4 (79.3; 85.2) | 160,079 | 8.8 (6.7; 11.4) | 160,273 | 8.8 (7.0; 11.0) |
| **Type of health care provided (assessed need)** | | | | | | |
| Disease prevention and health promotion | 94,144 | 83.0 (71.4; 90.5) | 10,584 | 9.3 (4.4; 18.8) | 8,720 | 7.7 (3.0; 18.1) |
| Consultations and procedures | 15,846,916 | 84.9 (83.9; 85.8) | 1,480,692 | 7.9 (7.2; 8.7) | 1,348,442 | 7.2 (6.6; 7.9) |
| Medium or high complexity service | 1,267,168 | 87.3 (85.0; 89.3) | 76,242 | 5.3 (4.0; 6.9) | 108,301 | 7.5 (6.0; 9.3) |
| Others | 186,271 | 88.3 (82.0; 92.6) | 15,628 | 7.4 (3.9; 13.5) | 9,067 | 4.3 (2.5; 7.4) |

n—absolute frequency

%—relative frequency

**Table 2. Raw analysis of access to prescription medicines in the Brazilian population using multinomial logistic regression.**

| | Access to prescription medicines | | | |
| | Yes, some | | No, none | |
| | OR (IC 95%) | p | OR (IC 95%) | p |
|---|---|---|---|---|
| *Predisposing factors* | | | | |
| **Age (years)** | | | | |
| 0–14 | 0.99 (0.85; 1.15) | 0.961 | 0.99 (0.85; 1.16) | 0.998 |
| 15–19 | 1.03 (0.81; 1.31) | 0.784 | 1.48 (1.19; 1.84) | < 0.001 |
| 20–39 | 0.97 (0.84; 1.12) | 0.723 | 1.19 (1.03; 1.37) | 0.013 |
| 40–59 | 1.36 (1.20; 1.55) | < 0.001 | 1.36 (1.19; 1.55) | < 0.001 |
| $\geq 60$ | 1 | | 1 | |
| **Sex** | | | | |
| Women | 1.19 (1.08; 1.31) | < 0.001 | 1.22 (1.10; 1.35) | < 0.001 |
| Men | 1 | | 1 | |
| **Level of education** | | | | |
| No schooling | 2.50 (2.00; 3.12) | < 0.001 | 1.67 (1.35; 2.06) | < 0.001 |
| Incomplete middle school | 2.43 (2.01; 2.94) | < 0.001 | 1.57 (1.31; 1.87) | < 0.001 |
| Complete middle school | 2.47 (1.99; 3.06) | < 0.001 | 1.82 (1.49; 2.22) | < 0.001 |
| Complete high school | 1.84 (1.50; 2.25) | < 0.001 | 1.57 (1.31; 1.89) | < 0.001 |
| Higher education | 1 | | 1 | |
| *Enabling factors* | | | | |
| **Family income per capita** | | | | |
| Up to $252.52 | 2.85 (2.51; 3.24) | < 0.001 | 1.88 (1.67; 2.12) | < 0.001 |
| $252.53 to $547.57 | 2.16 (1.90; 2.47) | < 0.001 | 1.40 (1.24; 1.59) | < 0.001 |
| $547.58 or more | 1 | | 1 | |
| **Nature of care** | | | | |
| Private | 1 | | 1 | |
| Public | 3.57 (3.13; 4.07) | < 0.001 | 1.69 (1.52; 1.89) | < 0.001 |
| Do not know / Do not remember | 2.05 (0.63; 6.67) | 0.232 | 1.56 (0.55; 4.39) | 0.393 |
| **Health insurance** | | | | |
| Yes | 1 | | 1 | |
| No | 3.06 (2.63; 3.54) | < 0.001 | 1.70 (1.50;1.92) | < 0.001 |
| **Paid appointment** | | | | |
| Yes | 1 | | 1 | |
| No | 2.50 (2.11; 2.98) | < 0.001 | 1.37 (1.19; 1.58) | < 0.001 |
| **Appointment via SUS** | | | | |
| Yes | 1 | | 1 | |
| No | 3.37 (2.97; 3.83) | < 0.001 | 1.67 (1.50; 1.85) | < 0.001 |
| Do not know / Do not remember | 2.29 (0.82; 6.41) | 0.113 | 0.68 (0.16; 2.83) | 0.601 |
| *Needed factors* | | | | |
| **General health status** | | | | |
| Very good or good | 1 | | 1 | |
| Regular | 1.68 (1.51; 1.86) | < 0.001 | 1.24 (1.12; 1.38) | < 0.001 |
| Bad or very bad | 2.63 (2.32; 2.98) | < 0.001 | 1.53 (1.33; 1.75) | < 0.001 |
| **Reason to seek health service (perceived need)** | | | | |
| Disease diagnosis and treatment | 1 | | 1 | |
| Disease prevention and health promotion | 0.85 (0.74; 0.99) | 0.041 | 1.26 (1.10; 1.44) | 0.001 |
| Consultations and procedures | 0.83 (0.0; 0.98) | 0.032 | 1.07 (0.91; 1.26) | 0.386 |

(*Continued*)

**Table 2.** (Continued)

| | Access to prescription medicines | | | |
|---|---|---|---|---|
| | **Yes, some** | | **No, none** | |
| | **OR (IC 95%)** | **p** | **OR (IC 95%)** | **p** |
| External causes and rehabilitation | 0.96 (0.81; 1.14) | 0.693 | 1.17 (90.99; 138) | 0.059 |
| **Type of health care provided (evaluated need)** | | | | |
| Disease prevention and health promotion | 1 | | 1 | |
| Consultation and procedures | 1.03 (0.59; 1.79) | 0.901 | 0.84 (0.50; 1.42) | 0.535 |
| Medium or high complexity service | 0.80 (0.45; 1.43) | 0.466 | 0.87 (0.50; 1.50) | 0.623 |
| Others | 0.99 (0.49; 2.01) | 0.990 | 0.95 (0.48; 1.86) | 0.891 |

OR–odds ratio

IC—confidence interval

p–p value

## Discussion

This study verified the access to medicines and associated factors in Brazil, considering the Andersen Behavior Model [13]. Our results showed individual characteristics influencing access to medicines and classified as predisposing (age, sex, and level of education), enabling (family income and nature of care), and needed factors (general health status and reason to seek health care).

Demographic structure, socioeconomic, behavioral, and cultural factors, morbidity profile, pharmaceutical market, and public policies influence access to medicines. A study on the American health system [13] confirmed that these factors directly influence access to essential medicines, as observed in Brazil [12]. Therefore, investigating them clarifies the role of medication in health, especially in the public health context [15].

Regarding associated factors, the present study found a higher prevalence of partial or no access to medicines among adults (40 to 59 years), females, less schooled, and lower-income individuals. Regarding enabling and needed factors, we found less access to medicines among individuals who received care in the public system (SUS), those with poor or very poor self-assessed health, and those who sought rehabilitation or received disease prevention services. Our findings highlight the social and economic inequity in access to medicines with the lack of access for the Brazilian population depending on the public health system and in more need of health care [4, 16, 17].

The National Research on Access, Utilization, and Promotion of Rational Use of Medicines (PNAUM) was conducted in Brazil in 2014. It showed that the lowest prevalence of access to medicines occurred among men and lower-income individuals, also indicating difficulties in access by individuals with chronic diseases [3, 18].

Access to essential medicines is related to vulnerable groups who need the public health system but usually only seek health services when they are sick or report a bad or very bad state of illness. Therefore, they seek prescription medicines to solve immediate health demands and prevent specific diseases or situations, such as prenatal care, childcare, or medical check-ups [18–20].

The bivariate analysis showed a greater probability of partial or no access to prescription medicines in the public system, indicating that public policies remain ineffective. Although the Brazilian government invested in policies to expand access to medicines through regulating generic medicines and the Popular Pharmacy Program, access is still limited. Resistance from

**Table 3. Adjusted analysis of access to prescription medicines in the Brazilian population using multinomial logistic regression.**

| | Access to prescription medicines | | | |
| --- | --- | --- | --- | --- |
| | Yes, some | | No, none | |
| | OR (IC 95%) | *p* | OR (IC 95%) | *p* |
| *Predisposing factors* | | | | |
| **Age (years)** | | | | |
| 0–14 | 1.21 (0.99; 1.47) | 0.055 | 1.00 (0.81; 1.24) | 0.931 |
| 15–19 | 0.94 (0.72; 1.22) | 0.657 | 1.28 (1.00; 1.63) | 0.043 |
| 20–39 | 1.00 (0.85; 1.19) | 0.922 | 1.14 (0.96; 1.34) | 0.113 |
| 40–59 | 1.30 (1.14; 1.49) | < 0.001 | 1.30 (1.13; 1.49) | < 0.001 |
| ≥ 60 | 1 | | 1 | |
| **Sex** | | | | |
| Women | 1.12 (1.01; 1.24) | 0.028 | 1.18 (1.06; 1.32) | 0.001 |
| Men | 1 | | 1 | |
| **Level of education** | | | | |
| No schooling | 1.03 (0.79; 1.33) | 0.815 | 1.24 (0.97; 1.59) | 0.081 |
| Incomplete middle school | 1.01 (0.81; 1.27) | 0.875 | 1.12 (0.91; 1.38) | 0.269 |
| Complete middle school | 1.20 (0.95; 1.53) | 0.121 | 1.26 (1.01; 1.59) | 0.040 |
| Complete high school | 1.09 (0.87; 1.35) | 0.432 | 1.24 (1.02; 1.50) | 0.028 |
| Higher education | 1 | | 1 | |
| *Enabling factors* | | | | |
| **Family income per capita** | | | | |
| Up to $252.52 | 1.67 (1.43; 196) | < 0.001 | 1.56 (1.34; 1.82) | < 0.001 |
| $252.53 to $547.57 | 1.44 (1.24; 1.67) | < 0.001 | 1.18 (1.02; 1.37) | 0.020 |
| $547.58 or more | 1 | | 1 | |
| **Nature of care** | | | | |
| Private | 1 | | 1 | |
| Public | 2.60 (2.23; 3.05) | < 0.001 | 1.36 (1.19; 1.55) | < 0.001 |
| Do not know / Do not remember | 1.76 (0.52; 5.95) | 0.361 | 1.04 (0.31; 3.43) | 0.940 |
| *Needed factors* | | | | |
| **General health status** | | | | |
| Very good or good | 1 | | 1 | |
| Regular | 1.43 (1.26; 1.61) | < 0.001 | 1.17 (1.04; 1.31) | 0.008 |
| Bad or very bad | 2.07 (1.79; 2.41) | < 0.001 | 1.36 (1.17; 1.59) | < 0.001 |
| **Reason to seek health service (perceived need)** | | | | |
| Disease diagnosis and treatment | 1 | | 1 | |
| Disease prevention and health promotion | 1.09 (0.93; 1.27) | 0.271 | 1.35 (1.17; 1.56) | < 0.001 |
| Consultation and procedures | 0.91 (0.76; 1.09) | 0.325 | 1.09 (0.92; 1.29) | 0.291 |
| External causes and rehabilitation | 1.10 (0.93; 1.31) | 0.244 | 1.22 (1.03; 1.45) | 0.018 |

LR chi2 = 837.64; $p < 0.001$; Pseudo $R^2$ = 0,034

OR–odds ratio

IC—confidence interval

p–p value

prescribing professionals and low acceptance of generic medicines in public and private sectors may also contribute to poor access to medicines. Moreover, after 2016, reduced resources compromised the Popular Pharmacy Program, leading to the extinction of the governmental pharmacy network and favoring the continuation of the program *Aqui tem Farmácia Popular*, linked to private networks of community pharmacies [21, 22].

PNAUM also showed greater access to medicines and pharmaceutical care in the public health system. Results showed greater access to medicines when a Procurement Committee was responsible for medication acquisition. This finding reinforces having a committee to organize and achieve pharmaceutical care goals. The Pharmacy and Therapeutics Committee and the National List of Essential Medicines (RENAME) are also strategic structures for pharmaceutical care [23]. Lack, shortage, or discontinuity of medication can strongly compromise access to health services. Therefore, in public health services, mainly in primary care, partial or no access to medicines impairs pharmaceutical care [7, 24].

In contrast, no availability of medicines due to insufficient funding and issues with local pharmaceutical care management may increase the prescription of non-essential medicines, decreasing the access to prescription medicines in pharmacies linked to the public health system. Thus, a periodical update in available medicines is needed to increase treatment adherence. The update is usually conducted by a multidisciplinary committee that must disseminate the list to prescribing professionals and maintain the regular pharmacy medication supply [25].

The multilevel analysis showed that women were more likely to have partial or no access to medication. Women often seek more health services than men in public and private systems. Also, the number of women seeking care in the last two weeks increased in the 2019 PNS compared to the 2013 edition. However, the overall use of health services was higher among men [20]. In general, women have better health self-perception and seek health promotion, disease prevention (e.g., cytological exam), and prenatal care. Meanwhile, men access and use emergency services more frequently, possibly explaining the high prevalence of appointments among them. Also, because they often only seek health services in sickness, the service culminates in medication use.

Contrarily, services accessed by women do not always result in medicine prescriptions, as they may be more preventive than curative. However, this study considered access to medicines based on prescription medicines. One explanation for our findings may be linked to the social role of women as primary responsible for the family, leading to them prioritizing more other expenses than their health, as observed in social policies during the COVID-19 pandemic. Furthermore, women may self-medicate with non-prescribed medicines, not requiring the prescription, either in public or private pharmacies [26].

Regarding education and income, individuals with complete middle or high school and from lower-income families presented more partial or no access to medicines than individuals with higher education. Our results corroborate the literature since population groups with higher age, family income, and education showed the highest prevalence of access to medicines. Older individuals use more continuous medication, especially to treat chronic non-communicable diseases. Those with a higher level of education may be more aware and responsible regarding health with more knowledge about the use of prescription medicines and the acceptability of access. Last, families with more purchasing power have easier access to prescription medicines according to their health needs [16, 27].

We also found a lower probability of access to prescription medicines in the public system. Drummond et al. [28] analyzed data from the 2013 PNS, showing that, despite the high prevalence of access to prescription medicines in Brazil, most individuals paid some amount for the medication, and only 15.3% had full access to prescription medicines in SUS. Moreover, trend analysis studies observed a decrease in individuals accessing medicines via SUS since 2008 [29–31]. Specifically, Boing and other authors observed that the proportion of people who did not obtain no medication in the SUS and that made some direct disbursement increased, comparing the results of the PNS between 2013 and 2019, and that the probability of obtaining all the medications in the SUS was higher among the poorest [31].

SUS is facing difficulties as a universal coverage system due to current fiscal austerity Brazilian policies, affecting medication research, development, acquisition, and availability in the public health system [29, 32]. The contradiction between the redistributive model in the Federal Constitution of 1988 and the low levels of public spending on health (including the production of goods and supplies) worsened after the Constitutional Amendment 95/2016, which froze public spending for 20 years. Along with other measures, such as reforms on social security and labor and the Constitutional Amendment 109/2021, the entire public system has been dismantled [33].

Shortage and discontinuity of medication due to demand (seasonality effects) or supply (crisis in national or international pharmaceutical industries) is another important factor in the public health system. It may reflect management failures in government, whether by lack of training, programming, or alternatives, including financial ones, to avoid shortages. Discontinuity in medication supply causes low availability at dispensing points and reduces access to medicines [34, 35].

Health self-assessment was directly associated with access to prescription medicines. The negative perception about general health status does not guarantee access to medicines once prescribed. This finding points to the stigmatization of medication use, i.e., the attitude of not using a prescribed medicine due to poor acceptance of the health condition, fear of social judgment, or fear of dependence due to continuous use. This attitude is recurrent in adult and older populations, which are age groups with more chronic non-communicable diseases. Therefore, although they acknowledge their health needs, they resist treatment, especially pharmacotherapy [36].

Regarding perceived need, no disease signs or symptoms are usually evident when disease prevention (prenatal care, childcare, or medical check-up) requires a medicine prescription. The lack of symptoms, the unavailability of the health service, high prices, and geographical limitations (distant dispensing unit) may discourage the acquisition and use of a certain medication [37].

Our study used a representative sample of the Brazilian population, revealing the main factors associated with access to medicines in the country. However, potential recall bias can be a limitation due to self-reported data, especially about medication acquisition. Limitations of cross-sectional studies, such as incidence/prevalence bias, possibly occurred. Also, associations can be studied, but causality cannot be attributed to these associations. In addition, data on access to medicines were collected from data on the use of health services, revealing only access to prescription medicines, thus subjective issues related to lack of access to medicines were not discussed. Therefore, future studies should include issues related to access to non-prescription medication and access locations.

The results indicate social and economic inequalities in access to medicines and cultural issues related to the individual perception of health. Our findings reveal health inequities and highlight the need to update public policies on medication and pharmaceutical care. Strengthening existing policies (generic drugs, the program *Aqui tem Farmácia Popular*, and update the RENAME) and valuing them throughout the health care system by managers and health professionals is also needed.

## Conclusions

Access to medicines among the Brazilian population is associated with social, economic, and health perception factors, highlighting the need to update existing public policies and encourage new policies to strengthen medication availability in health care, especially in primary care (gateway to SUS).

Factors associated with access to medicines are age, sex, level of education, family income, nature of care, general health status, and reason to seek health care. Pharmaceutical care is essential to ensure access to medicines as it organizes the flow of medication in public or private health systems. Pharmaceutical care is an interprofessional activity involving several professionals linked to the medication cycle, including managers, doctors, pharmacists, nurses, and technicians. A comprehensive reorganization of pharmaceutical care throughout the Brazilian health system is needed to optimize the selection, acquisition, and distribution of medicines. It may ensure the creation of an updated medicines list consistent with the epidemiological polarization in Brazil, contributing to the universalization of health care and reduction of health inequities.

## Acknowledgments

The authors acknowledge Probatus Academic Services for providing scientific language review and editing.

## Author Contributions

**Conceptualization:** Adriana Amorim de Farias Leal, Maria Helena Rodrigues Galvão, Arthur de Almeida Medeiros, Ângelo Giuseppe Roncalli.

**Formal analysis:** Adriana Amorim de Farias Leal, Maria Helena Rodrigues Galvão, Arthur de Almeida Medeiros.

**Methodology:** Adriana Amorim de Farias Leal, Maria Helena Rodrigues Galvão, Arthur de Almeida Medeiros.

**Project administration:** Ângelo Giuseppe Roncalli.

**Writing – original draft:** Adriana Amorim de Farias Leal, Maria Helena Rodrigues Galvão, Arthur de Almeida Medeiros.

**Writing – review & editing:** Adriana Amorim de Farias Leal, Maria Helena Rodrigues Galvão, Arthur de Almeida Medeiros, Ângelo Giuseppe Roncalli.

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
