## [Decision Letter · Decision Letter 0]

6 Nov 2022

PONE-D-22-26707Access to medicines among the Brazilian population based on the 2019 National Health SurveyPLOS ONE

Dear Adriana Amorim de Farias Leal, 

Thank you for submitting your manuscript to PLOS ONE. After careful consideration, we feel that it has merit and we invite you to submit a revised version, on specific issues, of the manuscript that addresses the points raised during the review process. The objective, the design and the results of this study are relevant. 

We look forward to receiving your revised manuscript.

Kind regards,

Fernanda Penido Matozinhos, Ph.D

Academic Editor

PLOS ONE

Journal Requirements:

Additional Editor Comments:

Dear Adriana Amorim de Farias Leal,

Thank you for submitting your manuscript to PLOS ONE. After careful consideration, we feel that it has merit and we invite you to submit a revised version, on specific issues, of the manuscript that addresses the points raised during the review process. The objective, the design and the results of this study are relevant.

Reviewers' comments:

Reviewer's Responses to Questions

**Comments to the Author**

1. Is the manuscript technically sound, and do the data support the conclusions?

Reviewer #1: Yes

Reviewer #2: Yes

2. Has the statistical analysis been performed appropriately and rigorously? 

Reviewer #1: Yes

Reviewer #2: Yes

3. Have the authors made all data underlying the findings in their manuscript fully available?

Reviewer #1: Yes

Reviewer #2: Yes

4. Is the manuscript presented in an intelligible fashion and written in standard English?

Reviewer #1: Yes

Reviewer #2: Yes

5. Review Comments to the Author

Reviewer #1: We are grateful for the opportunity to review this manuscript which aims to assess access to pharmacological treatments in the Brazilian population and the factors associated with this accessibility.

The objective is relevant and the results are of interest both for the setting in which the study is conducted and for a wide range of potential readers.

The design is convenient, it uses a familiar conceptual framework, the Andersen behavioural model, and the analysis methodology seems appropriate to the nature of the data.

However, we believe that the authors should address some recommendations in order to improve the manuscript.

In the Introduction section, we missed a slightly broader description of the framework in which health care is provided in Brazil, the organisation of its health care system, who is covered and how drugs are paid for at these different levels. Some of the considerations in the Discussion section would be better understood with this background information.

In the Methods section, the description of the dependent variable needs to be rewritten. Why is it stated that only affirmative answers to the question were considered? It would be valuable to attach information (maybe an annex) with the specific PNS questions that led to the definition of the variables. Since it is noted that the descriptive analysis considered the sample weight and the effect of clustering, the authors should show which method was used to estimate the confidence intervals. In this section one would also expect to find information on how the best model was chosen and whether any measures of fit were assessed.

The results section needs additional information. What does column n in table 1 mean?

The choice of multilevel models, which is quite appropriate in this case, opens the door to examine information of interest to readers. To assess the relevance of clustering into specific geographical units (e.g. districts) in explaining access to drug treatments, multilevel variability measures can be used. We recommend the approach proposed by Merlo et al (J Epidemiol Community Health. 2006 Apr; 60(4): 290-297.) The median odds ratio (MOR), which can be understood as the increase in risk you would have (at the median) if you were to move to another area with a higher risk, could be estimated and reported.

In explaining the results of the multivariate analysis, it should be noted that the excess risk needs to be interpreted in relation to the reference category (having full access to prescribed medicines). This may seem obvious to the authors, but will not be so for readers unfamiliar with multinomial logistic regression.

The discussion is sufficiently broad but could perhaps be reworded to make it easier to follow. In the third paragraph it is noted "Regarding associated factors, the present study found a higher prevalence of partial or no access to medicines among adults (40 to 59 years), males, less schooled, and lower-income individuals. We believe that there must be an error as the risk of being in these categories is higher for women (table 3). This error is also shown in the results of the abstract.

We do not know whether subjective reasons for lack of access to medication were asked. In the discussion section, the different reasons are considered from a theoretical point of view. If this was not done, it should be presented as a limitation of the study. Another type of limitation is associated with the study design (cross-sectional). Associations can be studied, but causality cannot be attributed to these associations. Therefore, the first conclusion cannot be derived from the results of the study. The way the conclusions are presented in the abstract seems more appropriate.

In conclusion, it appears to be a study that deserves to be published but which would benefit from a certain degree of revision, on specific issues, on the way in which the results are communicated, and on the assessment of the scope of the findings presented.

Reviewer #2: A recently published article analysed data from the same data source (PNS) comparing 2013 and 2019, coinciding with the time frame of present manuscript, but no discussion is provided on how the data compare and if there are significant differences in the results .

Reference 31 - Boing AC, Andrade FB, Bertoldi AD, Peres KGA, Massuda A, Boing AF. Prevalence

rates and inequalities in access to medicines by users of the Brazilian Unified National Health

System in 2013 and 2019. Cad Saude Publica. 2022;38(6):e00114721. doi: 10.1590/0102-

311xpt114721

6. PLOS authors have the option to publish the peer review history of their article (what does this mean?). If published, this will include your full peer review and any attached files.

Reviewer #1: **Yes: **Jesús Martín-Fernández

Reviewer #2: No

---

## [Author Response · Author response to Decision Letter 0]

29 Dec 2022

Was inserted in the Introduction the description of the framework in which health care is provided in Brazil and the reference has been updated. In the methods, the questions from the PNS 2019 that led to the definition of the sample were inserted, qnd was explained the analysis of the fit of the final model was performed using the likelihood ratio test and Pseudo R2. The reference category (having full access to prescribed medication) was informed in the results. The error in the Results section on gender has been fixed. The beginning of the conclusion was adjusted and the data from the study by Boing et al. (reference 31) were discussed.

---

## [Editor Report · Decision Letter 1]

4 Jan 2023

Access to medicines among the Brazilian population based on the 2019 National Health Survey

PONE-D-22-26707R1

Dear,

We’re pleased to inform you that your manuscript has been judged scientifically suitable for publication and will be formally accepted for publication once it meets all outstanding technical requirements.

Kind regards,

Fernanda Penido Matozinhos, Ph.D

Academic Editor

PLOS ONE

Additional Editor Comments (optional):

Dear Author,

Thank you for the opportunity to review this manuscript. I am grateful for the invitation.

After careful consideration, I feel the manuscript explores a very important topic. The questions were responded and modifications in the text made the manuscript come to a satisfying result.

Kind regards,
---

## [Editor Report · Acceptance letter]

11 Jan 2023

PONE-D-22-26707R1 

Access to medicines among the Brazilian population based on the 2019 National Health Survey 

Dear Dr. Leal:

I'm pleased to inform you that your manuscript has been deemed suitable for publication in PLOS ONE. Congratulations! Your manuscript is now with our production department. 

Kind regards, 

on behalf of

Dr. Fernanda Penido Matozinhos 

Academic Editor

PLOS ONE